# Research on the Irrational Behavior of Consumers’ Safe Consumption and Its Influencing Factors

**DOI:** 10.3390/ijerph15122764

**Published:** 2018-12-06

**Authors:** Jianhua Wang, Minmin Shen, Ziqiu Gao

**Affiliations:** 1School of Business, Jiangnan University, Wuxi 214122, China; 6180909014@stu.jiangnan.edu.cn (M.S.); 6180909007@stu.jiangnan.edu.cn (Z.G.); 2Jiangsu Research Center of Food Safety, Jiangnan University, Wuxi 214122, China

**Keywords:** safe consumption, irrational behavior, safety-certified pork, purchase intention, consumer preference

## Abstract

Frequent food safety incidents in recent years have greatly reduced consumers’ trust, and consumers’ demand for safe food has been on the rise. However, there is an inconsistency between the consumers’ willingness and actual purchasing behaviors. Some consumers who have a willingness to purchase safe food ultimately do not produce actual purchasing behaviors, resulting in an “irrational behavior” in the safe food consumer market. In order to better study this phenomenon and identify its inherent logic, we chose to use pork (a typical representative of safety-certified agricultural products) as the object, based on a survey on 844 consumers in the Jiangsu Province and Anhui Province analyzed in July 2017 by RPL (Random Parameters Logit) and binary Logit regression methods from two aspects, i.e. consumer preference for different attributes of safety-certified products and factors affecting safe consumption. The research results show that consumers have a significant preference for pork that has additional attributes such as green food certification, organic food certification, origin information and “No Additives and Veterinary Drug Residue Labeling”; labeling such information on the pork can effectively improve consumers’ trust. Consumers’ inconsistency of purchase intention with purchasing behaviors of safety-certified pork is affected by many factors, such as gender, age, annual household income, the degree of trust in agricultural product quality and a safety certification mark, understanding of safety-certified pork, and the degree of concern on pork quality and safety issues. These factors have all contributed, to varying degrees, to the rising of “irrational behavior” of consumers’ safe consumption, lead to an irrational state of consumption that consumers with a safely certified pork purchase will not necessarily buy a safety-certified pork. Based on the results of two empirical analyses, it can be concluded that pricing and age are the two main influencing factors that lead to the “irrational behavior” of consumers’ safe consumption.

## 1. Introduction

Over 200 diseases are caused by unsafe food containing harmful bacteria, parasites, viruses, and chemical substances, and it is estimated that two million deaths occur every year from contaminated food or drinking water [1]. Food safety has become a hot topic in the world that needs to be addressed urgently by all governments. Although the overall food quality and safety level in China has been steadily rising in recent years, frequent food safety incidents at home and abroad have led to more consumer concerns in food safety in their purchasing behaviors. The consumers’ awareness of acquiring and tracking food information has also gradually increased [2].

Safe food should be a product that does not have the potential to harm or threaten human health, including the safety of the production, operations, results, and process, as well as of the reality and future [3]. However, as a safe food production and supply chain is yet put in place, the information of both sides of the transaction is asymmetrical, and the government’s supervision and punishment system is yet to be perfect, artificial risk still exist in the Chinese food market [4], such as speculative behavior because of market regulatory loopholes (in 2004, tens of thousands of pork from diseased or dead pigs in Jiangxi Province were sold to 7 provinces such as Guangdong, Hunan, Henan, and Anhui), leading to the result that even the consumers who have the willingness to purchase safe food may not actually buy safe food. This is the reason behind the irrational behavior of safe consumption in the market, consumers’ strong potential demand for food has not changed into a reality, which creates a dilemma in which safe food is unsellable on one hand, and on the other hand, it is difficult for consumers to find safe food. In the frequent food safety incidents, those caused by meat products accounted for the highest proportion, and in meat food safety accidents, the pork safety issue was the first to bear the brunt and occurred most frequently [5]. Pork safety problems occur in many links of the pork supply chain, including the excessive use of veterinary drugs and veterinary antibiotics in pig breeding, illegal water injections in slaughtering, using clenbuterol or other substances, “illegal smuggling” in transportation and sales, and the processing and sale of “expired and degraded pork”. Therefore, pork was included in the first batch of certified products to regulate the production and supply of pork [6]. In view of this, this paper selects pork as a typical safety-certified agricultural product as a research area into consumers’ willingness to consume safely and purchasing behavior from two aspects, i.e., different attributes of safety-certified products and factors affecting consumer safe consumption, which is of great practical significance and reference value to establish a sound safety-certified food market that suits China’s national conditions. 

## 2. Literature Review

Food production in China is a mostly decentralized operation of small workshops, with simple production equipment and backward production technology, so it is difficult for the government and the market to effectively restrain it. The consumers, due to factors such as information asymmetry and a low degree of trust, consumers who have a willingness to purchase safety-certified pork may also generate irrational purchasing behavior. Irrational consumption generally occurs in the situation when consumers consume without logical reasons or clear thinking due to little knowledge and understanding of the product. Daniel Kahnema, an American expert in economics and psychology believes that human irrational consumption behavior will also affect the objective market [7]. Therefore, the irrational behavior of consumer’s safe consumption will have a certain impact on the safe food market. Consumers are demanders of safe food and also the only market subject able to test the value of safe food, their acceptance of safe food and willingness to consuming determine the development and future prospects of the safe food market. The establishment of the domestic safe food market requires a gradual process, and consumers also need a cognitive process for safe food [8]. An immature buyer’s market at the consumer end will also lead to the imbalance of the safe consumption market. At present, there is still a lack of trust of Chinese consumers on food quality and safety standards. Take green food for example. According to Zhang Liguo and Zeng Yinchu, most consumers have a relatively low understanding and trust of green food which, in turn, leads to the insufficient purchase of green food [9,10]. In this case, consumers are reluctant to pay an exorbitant premium for high-quality products. In Ye Yan’s research on organic tea, the market price of organic tea generally has a premium of more than 50% when compared with ordinary tea, but consumers are only willing to pay a premium of 39.96% for organic tea, making it difficult for organic tea producers to obtain reasonable economic returns, thus reducing the production enthusiasm [11], which is also one of the main reasons for the phenomenon of “bad money drives out good” in the safe food market. When the market fails, the safe food producers will bear high production costs, having to bear the ecological environment input cost that should have been borne by the society, but never getting the corresponding compensation. As a result, the producers’ willingness to produce safe food reduces, the supply of safety-certified agricultural products is insufficient, and the market finds it difficult to reach Pareto Optimality [12].

Consumers buy products mainly to meet their own needs [13]. Therefore, the inherent properties of food such as quality, taste, and nutritional value are the main factors consumers consider. The external features of food, such as its brand, price, and label, can help consumers identify the quality and value of food in the case of information asymmetry between the sides of the transaction and increase the possibility of consumer purchase [14]. The quality and safety certification of agricultural products is one of the main tools to prevent food safety risks. The “three products” certification (organic, green, and pollution-free) is an important indicator for assessing food quality and safety in China [15]. Ortega et al. (2016) studied the willingness of Chinese consumers to pay for the quality and safety attributes of beef. The results showed that consumers had a significant preference for green and organic certified beef and were willing to pay higher prices for green-certified beef [16]. The origin of food is also an important factor affecting consumers’ purchase intention. Wägeli et al. (2016) found that as consumers become more aware of the growing environment and production processes of locally produced food and the fact that short-distance transportation could keep food fresh, they prefer to buy local organic food [17]. The information on additives and drug residues contained in food is also an important influencing factor. In a choice experiment conducted by Zhang Zhen et al. (2013) on pork, it is found that consumers had an obvious purchasing preference for pork with quality assurance (excluding illegal additives such as “clenbuterol” and veterinary drug residues) [18]. 

Influenced by factors such as individual and family characteristics and the cognitive level of consumers, there are significant differences in the preferences of different consumers for product attributes. Research shows that consumer safe consumption is affected by influencing factors, including the consumers’ gender, age, education background, and annual household income [19]. According to a study by Smith et al. (2009), the consumer demand for milk is influenced by factors such as the milk price, substitute products’ prices, and the consumer’s milk safety cognition [20]. Aletkure et al. (2010) pointed out that gender and socioeconomic status significantly affect consumers’ safe food purchasing behavior. Compared with women, men are more inclined to choose to take risks to purchase products without safety certifications. As the socioeconomic status rises, risk preference will also decrease accordingly [21]. In 2011, Unklesbay conducted a survey of food safety awareness of 824 college students in the United States. The results showed that consumers’ knowledge of food safety significantly affected their safe consumption behavior. In the survey, students majoring in food science, nutrition, and health care were more inclined to purchase products with a safety certification, while students who had never studied food-related courses generally had a weak sense of safe consumption. 

To sum up, for the irrational phenomenon in consumer safety consumption, despite the influence of national conditions and culture, consumers generally attached importance to safety certification attributes, origin information, quality assurance labels, etc., and the purchase preferences for these attributes or combinations of attributes. Consumers’ individual characteristics, family characteristics, and marketing activities all have an impact on consumers’ safe consumption. However, domestic and foreign research has focused on the overall status quo and existing problems, and there are few interpretations from the perspective of consumers. In addition, in the international study of the choice of experimental methods to study consumer preferences for food attributes, whether the attributes and level settings are in line with China’s national conditions remains to be seen. Therefore, this paper selects the certification level, origin, quality, and safety assurance attributes of ham butt to attempt to determine the safety-certified pork attributes combinations and price positioning that meet both consumer preferences and consumer utility needs. The factors influencing consumers’ safe consumption were discussed from basic individual characteristics including their gender, age, education background, and understanding and cognition of pork safety, and basic individual characteristics including annual household income and whether there were children under 18 years old. 

## 3. Theoretical Basis and Modeling

The theoretical basis of this paper uses the Discrete Choice Theory and Lancaster Consumer Theory. Discrete Choice Theory was proposed by McFadden in 1974 to study how consumers allocate their income to meet their needs to the greatest extent when paying for various goods and services, an important theory in consumer choice and demand research. Discrete Choice Theory holds that several attributes constitute a product or a service, and the specific utility of these attributes can be observed through instance variables. Consumers have different attitudes and cognitions towards different attributes of products or services, and then give different assessments, that is, part-worth utilities of the attributes. After the consumer comprehensively evaluates the attribute utilities of a product or service, the resulting overall evaluation of the product or service is referred to as the overall utilities. Several studies have shown that the assessed value of overall utilities can significantly influence whether consumers will purchase such products or services [22]. On the contrary, Lancaster breaks through the research limitations of traditional economics and proposes that consumers do not get utility directly from the product itself, but that the product’s attributive characteristics meet a certain demand, that is, the utility of the product comes from the attributes of the product rather than the product itself. The purchasing behavior of consumers is, in essence, to select products with different attributes and attribute hierarchies. Lancaster’s theory mainly includes the following three assumptions: (1) it is not the product that brings utility value to consumers, but the attributes and attribute hierarchies of the product; (2) a product may have one or more attributes and attribute hierarchies, and the same attribute or attribute hierarchy may be jointly owned by multiple products; (3) different attributes and attribute hierarchies are permuted and combined to form different products, and the attributive characters of a single product and composite product are also different [23]. 

Therefore, in this paper, the safety-certified pork is defined as a product profile of various safety-certified pork products, which are randomly combined by various attributes and attribute hierarchies, including green food certification, pollution-free agricultural product certification, organic food certification, “No Additives and Veterinary Drug Residue” label, origin information and price. This paper mainly adopts Lancaster Consumer Theory and studies how consumers select the safety-certified pork categories with different attribute combinations under the restriction of their own income to realize the maximization of their own utility.

It is now assumed that $Y_{nik}$ represents the utility of the experimental participant *n* in selecting the *i*-th safety-certified pork contour profile from the subset *v* of set U in *k* scenarios. However, the consumption utility of consumer safety-certified pork is composed of a deterministic part $X_{nik}\;$and stochastic part $\varepsilon_{nik}$, and the formula is as follows:(1)$$Y_{nik}=X_{nik}+\varepsilon_{nik}$$

Only if $Y_{nik}>Y_{njk}$, that is, $X_{nik}-X_{njk}>\varepsilon_{nik}-\varepsilon_{njk}$ holds for any *j* ≠ *i*, will the experimental participant *n* choose the *i*-th safety-certified pork profile, and the probability that consumers choose the *i*-th safety-certified pork profile is
(2)$$P_{nik}=prob(X_{nik}-X_{njk}>\varepsilon_{nik}-\varepsilon_{njk};\forall j\neq i)$$

$X_{nik}$ represents linear functions of different attribute categories, expressed as follows:(3)$$X_{nik}=a_{n}\beta_{nik}$$

$a_{n}$ represents the score utility vector of the experimental participant *n*, and $\beta_{nik}\;$represents the attribute vector of the *i*-th safety-certified pork in the *k*-th scenario.

When analyzing multiple repeated choices made by the same experimental participant, the Multinominal Logit (MNL) model and Random Parameters Logit (RPL) model shall apply. However, the MNL model requires the homogeneity of the experimenter’s preference, which is not consistent with the hypothesis of this paper, while the RPL model allows heterogeneity in the experimenter’s preference, so this paper selects the RPL model for the related research.

This paper further assumes that $\varepsilon_{nik}$ follows the extreme value distribution of type 1, then the probability that the experimental participant n chooses the safety-certified pork *i* attribute in the *k* scenario can be expressed as
(4)$$P_{nik}=\int \frac{\exp\left(X_{nik}\right)}{\sum_{j}\exp\left(X_{nik}\right)}f\left(b_{n}\right)d_{b_{n}}$$
where, *f*(*b*) is the probability density function of parameter *b*. When *f*(*b*) is a discrete function, the above formula can be further changed into the Latent Class Model (LCM), and then the experimental participants with the same or similar preferences shall fall into the same category according to the matching degree of the model and the data. Eventually, all of the participants are divided into the *t* category. Additionally, the probability that the experimental participant n is divided into the *t*-th category and selects the *i*-th safety-certified pork profile is
(5)$$P_{nik}=\sum_{t=1}^{T}\frac{\exp\left(c_{t}\beta_{nik}\right)}{\sum_{j}\exp\left(c_{t}\beta_{nik}\right)}R_{nt}$$

In the above formula, $c_{t}\;$represents the parameter vector of the experimental participants’ group of the *t* category, and $R_{nt}$ represents the probability that the experimental participant *n* falls into the *t* category, which can be expressed as
(6)$$P_{nik}=\frac{\exp\left(d_{t}Z_{n}\right)}{\sum \exp\left(d_{t}\beta_{nik}\right)}$$

$Z_{n}$ represents a series of observed values affecting the experimental participant *n* in a certain category and $d_{t}\;$is a parameter vector of experimental participants in the *t* category.

## 4. Experimental Design and Sample Analysis

### 4.1. Experimental Design

After summarizing the research conclusions of domestic and foreign scholars on safety-certified products, this paper chooses to conduct research on the basic attributes that consumers generally pay attention to in pork consumption, including certification level (no certification, pollution-free agricultural product certification, green food certification, organic food certification), origin information (without origin information or with origin information), quality and safety assurance label (without “No Additives and Veterinary Drug Residue Label” or with “No Additives and Veterinary Drug Residue Label”), and price (30 yuan/kg, 50 yuan/kg, 80 yuan/kg).

For the attributes and code settings of ham butt, please refer to Table 1 below.

According to the setting of the attribute and attribute hierarchy of safety-certified pork, (4 × 2 × 2 × 3) 48 kinds of virtual safety-certified pork product profiles can be formed. If all the safety-certified pork product profiles are combined, the full factorial design can be used to generate ((4 × 2 × 2 × 3)^2^) 2304 kinds of safety-certified pork profiles. Apparently, it is not realistic for consumers to make choices from 2304 product combinations. What is more, the average consumer will easily be fatigued after identifying more than 15 product profiles [24]. In order to ensure the experimenter’s selection efficiency and avoid strategic deviation, this paper adopts the experimental design module in the JMP11.0 software (SAS, Raleigh, NC, USA) and uses the fractional factorial design and the D-efficiency methods to ultimately determine the 12 different product combinations to form a questionnaire on the premise of the balanced distribution of various attributes and attribute hierarchies. Based on the research by Lusk et al., if the “No Option” is omitted and the two options are both unattractive, it may cause consumers to be forced to make unreal choices and distort their true preferences [25]. Therefore, this questionnaire consists of two different safety-certified pork selection sets and a “No Option” to obtain more realistic selection and improve the authenticity and effectiveness of the data.

The questionnaire survey was conducted in Jiangsu Province and Anhui Province. The two provinces are adjacent to each other and belong to Eastern China. The economic development levels of the two provinces are somewhat different, and their consumption levels and consumption habits are also different. Therefore, consumers in these two provinces can roughly characterize the preferences and purchase intentions of the Chinese consumers for different attributes and attribute hierarchies of safety-certified pork. Prior to the formal investigation, a pre-survey was conducted, followed by a summary and improvement of the questionnaire to delete poor questions through expert evaluation and internal test, and all enumerators had been trained to familiarize themselves with the questionnaire. In order to ensure the reasonable distribution of the samples and the authenticity and effectiveness of the data, this survey was conducted following the principle of the layered design in nine representative cities in the Jiangsu Province (South Jiangsu: Suzhou, Wuxi, Changzhou; Central Jiangsu: Nantong, Yangzhou, Taizhou; North Jiangsu: Huai’an, Suqian, Xuzhou) and three representative cities in the Anhui Province (South Anhui: Xuancheng; Central Anhui: Hefei; North Anhui: Bengbu) in July 2017. A trained investigator selects each city’s large-scale supermarket, shopping mall, large-scale farmer’s market, agricultural and sideline products stores, and other gathering places for meat consumers to conduct a 20–30 minute questionnaire survey on consumers with pork purchasing experience (the investigation involved the behavioral characteristics of consumers and does not include consumer privacy, and the results of the survey were only used for academic research and will not be disclosed). A total of 984 questionnaires were distributed in this survey, of which 140 invalid questionnaires were excluded for inconsistent questionnaire answers (the respondents chose the option that he or she did not purchase safety-certified pork, but still answered the questions about the considerations in the purchase process) and incomplete questionnaires, and 844 valid questionnaires were finally collected, including 475 valid questionnaires in the Jiangsu Province and 369 valid questionnaires in the Anhui Province. The effective rate of the questionnaires was 85.77%.

### 4.2. Sample Characteristic Analysis 

The personal characteristics statistics of the sample consumers are processed using an SPSS descriptive analysis (IBM, Armonk, NY, USA) and shown in Table 2. In terms of the gender structure, among the 844 valid questionnaires, there were 371 males and 473 females, accounting for 43.96% and 56.04% of the total samples respectively. The proportion of men and women was basically the same in the two provinces, while the proportion of women was slightly higher than that of men, which was also consistent with the fact that women in real life were more engaged in family buying activities. In terms of the age distribution, the respondents had a large age span, covering basically all the age groups, among which age 30 and below and age 40–49 accounted for the largest proportion, 30.21% and 26.42%, respectively. By analyzing the data of the sample consumers’ education background, it could be found that the respondents with no more than a high school education accounted for the largest proportion (29.38%) and those with a senior high school or technical secondary school education and undergraduate degree accounted for the same proportion, all reaching 26.07%. Since there was a majority of middle-aged and elderly consumers in the samples, more than half of the people whose education was below 9 years could be explained by the reality, while more than 30% of the respondents were aged 30 or below, so it was understandable that a bachelor degree accounted for a relatively large proportion. In terms of the annual household income, households with an annual income of more than 80,000 accounted for 61.37%, indicating that most of the households surveyed had higher incomes. This is because the survey site was in Eastern China, which has an economy that develops rapidly, and the people’s living standards are high, which are consistent with the survey data. Thus, families with high incomes have a better ability and the conditions to purchase safety-certified pork. In the survey samples, the data of whether there were minors under 18 in the family were basically flat, with the proportion of families without minors slightly higher than that of those with minors. 

The statistics of consumers’ cognition and consumption of safety-certified pork are processed in an SPSS descriptive analysis and shown in Table 3. The customers had a poor understanding of safety-certified pork. Only 17.65% of consumers knew a bit about the safety-certified pork, while only 1.78% had a good understanding of safety-certified pork. A total of 80.57% of consumers did not know about or felt strange regarding safety-certified pork. However, when it came to pork safety, most people were concerned or very concerned about it (68.01%). It could be seen that the current pork safety problem was serious and consumers were concerned about it. For the quality and safety certification mark of agricultural products, more than half of the consumers chose to trust and trust a lot, and nearly 80% of consumers held a positive attitude towards safety certification. A total of 91.59% of consumers said that they intended to purchase safety-certified pork, however, only 57.58% of consumers had purchased safety-certified pork, which showed that the proportion of the consistency of consumers’ self-reported preferences and realistic choices for purchasing safety-certified pork was very low and most consumers had very different self-reported preferences and realistic choices. 

## 5. Model Estimation and Discussion

### 5.1. Analysis of Consumer Preference on Different Attributes of Safety-Certified Pork

Based on the setting of the attribute and attribute hierarchy of safety-certified pork, the effect code is adopted in this paper to assign values to each attribute hierarchy, as shown in Table 4 below.

In this paper, the Hatlon algorithm in the Nlogit software (ESI, New York, NY, USA) was used to simulate and estimate the consumers’ utility value, so as to conduct a quantitative analysis of the preferences of different categories of attributes and attribute hierarchies of the safety-certified pork by different consumer groups.

According to the regression results of the Mixed Logit model in Table 5 below, in the main effect, in addition to the pollution-free agricultural product certification, Consumers’ preference for green food certification, organic food certification, origin information, and “No Additives and Veterinary Drug Residue Label” is significant at the 1% statistical level, which proves the existence of consumer heterogeneity and also proves that labelling the safety certification mark on pork, adding the origin information, and indicating the presence or absence of additives and veterinary drug residues can effectively improve consumers’ trust in pork safety, provide consumers with the weak side of the information asymmetry, with more knowledge for judgments, and that this has a positive impact on improving consumers’ purchase intention and purchasing behavior. Among the attributes of the certification level, compared with no authentication information, the coefficient of organic food certification is the largest, reaching 0.5393, followed by green food certification with a coefficient of 0.4111, which suggests that consumers show a greater preference for organic food certification, which may be related to the fact that organic food certification represents the highest level of food safety and also indicates from another side that consumers are more inclined to purchase pork with a higher safety factor. In addition, in the consumers’ purchase intention for safety-certified pork, the price factor is negatively significant at the 1% level. It can be seen that the price is an important consideration in the safe consumption of pork for consumers and that there is a general premium for safety-certified pork over ordinary pork so consumers may give up buying safety-certified pork for price reasons.

In the experiment verifying the interaction effect of safety-certified pork attributes, the interaction between annual household income, green food certification, origin information, and no additives and veterinary drug residue label were significant and significantly positive at the 1% statistical level, with the coefficients being 0.0861, 0.0737, and 0.0646, respectively, which indicated that high-income households preferred the pork with green food certification, origin information, and with “no additives and veterinary drug residue label”, and had the strongest preference for green food certification. The interaction effect between organic food certification and age was significantly negative at the 1% level, while the interaction effect between organic food certification and annual household income was significantly positive at the 5% level, which indicated that younger consumers and high-income families were more likely to purchase pork labeled with an organic food certification label. In the interaction between origin information and age, the interaction effect was significantly negative at the 10% level, indicating that young people would pay more attention to the origin information when purchasing pork than older consumers. In general, the cross-term effect of each attribute hierarchy with annual household income and age is significant, while the cross-term effect with gender and education is not significant. 

### 5.2. Analysis of the Influencing Factors of Irrational Behavior of Consumers’ Safe Consumption

In the questionnaire, the study sets two questions “Do you have the idea of purchasing safety-certified pork” and “Have you ever purchased safety-certified pork in your daily life?” to infer consumers’ willingness to purchase safety-certified pork and their purchasing behavior. For the convenience of experimental data processing, this paper will set the behavior of willing to purchase and with actual purchase as y = 1, and the behavior of willing to purchase but without actual purchase as y = 0, as a dependent variable for the Logit regression model analysis. 

This paper assumes that consumers’ purchase intention and actual purchasing behavior of the safety-certified pork are related to the consumers’ basic individual characteristics, family characteristics, consumers’ trust in the quality and safety certification mark of agricultural products, and consumers’ trust in and concern about the safety-certified pork. The gender, age, education background, annual household income, and whether there are children under the age of 18 in the household of consumers are all set as flag variables. According to Table 6, a binary Logit regression model is available: (7)LOG[P(Y1)P(Y2)]=β1gender+β2age+β3education+β4income+β5kid+εi

As can be seen from the results of model estimation in Table 7, consumers’ purchase intention and purchasing behavior of safety-certified pork are affected by many factors, among which the test value of gender and understanding of safety-certified pork is significant at the test level of 1%; age, annual household income, and concern about pork quality and safety are significant at 10% of the test level.

From the regression coefficient, for the influence on the consistency of consumers’ purchase intention and purchasing behavior for safety-certified pork, the estimated coefficients of age, annual household income, the trust in the quality and safety certification mark of agricultural products, the understanding of safety-certified pork, and the concern about quality and safety of pork are all positive, of which the consumers’ understanding of safety-certified pork has the greatest impact on the consistency of consumer safe consumption intention and behavior. When consumers’ understanding of safety-certified pork goes up to a new level, the possibility of their safe consumption consistency will increase by 61.4%, that is, when consumers intend to consume safely, the possibility of purchasing behavior will increase significantly, as does the trust in quality and safety certification mark of agricultural products, age, annual household income and concern about the quality and safety of pork, with increases of 22.7%, 16.2%, 13.8%, and 13.8%, respectively. It is worth noting that the estimated coefficient of gender’s influence on the consistency of consumers’ purchase intention and purchasing behavior is negative, which indicates that female consumers are more likely to put their willingness to purchase safety-certified pork into practice, while male consumers are less likely to turn their purchase intention into purchasing behavior. 

## 6. Discussion and Conclusions

Chinese scholars have done a lot of research on Chinese consumer’s safety food consumption behavior. Wang Zhigang conducted a survey on the main influencing factors affecting consumers’ food safety awareness and purchasing behavior in 2010. The results showed that consumers’ education background, living environment, smoking or not smoking, and the degree of concern about food safety had a significant impact on consumers’ awareness of safe food and purchasing behavior [26]. In the same year, Zhou Jiehong conducted research on the safety cognition and purchasing behavior of urban residents in Zhejiang Province, and the results showed that the educational background, family structure, consumers’ concern about vegetable safety and their cognition degree of vegetable safety standard were significantly correlated with consumers’ cognition of safe vegetable consumption [27]. This paper studies consumers’ willingness to consume and purchasing behavior of certified pork which is a typical safety-certified agricultural product based on a survey of 844 consumers in Massachusetts Jiangsu and Anhui provinces. The main conclusions are as follows: the proportion of the consistency of consumers’ self-reported preferences and realistic choices for purchasing safely certified pork is very low. The gap between self-reported preferences and realistic choices are wide for most consumers, leading to the “irrational behavior” of safe consumption. Additionally, labelling the safety certification mark on pork and adding the origin information and indicating the presence or absence of additives and veterinary drug residues can (1) effectively improve consumers’ trust in pork safety, (2) provide consumers, who are at the weak side of information asymmetry, with more grounds for making judgments, and (3) have a positive impact on improving both consumers’ purchase intention and their purchasing behavior. In addition, consumers’ inconsistency between their intention and behavior is affected by many factors. Consumers’ age, annual household income, the understanding of safety-certified pork, and the concern about quality and safety of pork, etc., all have a positive impact on the consistency of consumer safe consumption behaviors.

Accounting for the largest proportion of meat consumption in China and frequent pork safety issues, this study adopts safety-certified pork as the experimental subject to explore the influencing factors of consumers’ safe consumption and preferences over product safety attribute hierarchy, which is an area that previous scholars paid little attention to. At the same time, due to the convenience and geographical advantages, the survey locations were set up in Jiangsu Province and Anhui Province. Therefore, in future research, foods of other categories and data from other provinces in China, especially in the western region, can be tapped into to further complement and improve the existing research results. 

However, the above research conclusions also have important relevance to improving the safety-certified product markets of other categories. Firstly, the government needs to strengthen the supervision of food safety and increase the acceptance and trust of consumers on the safety-related certification. Secondly, it is feasible for the government to encourage producers to introduce safety certification, origin, and other information to safety food attributes. However, given the current government failure and market failure in the safe food market, the government should encourage the development of third-party certifiers in the efforts to standardize the food safety certification market and ensure healthy and orderly competition. Thirdly, due to the prevalence of a safe food premium, many consumers who have a willingness to conduct safe consumption are discouraged from doing so. The government can promote cooperation between manufacturers and higher institutions to strengthen the research and development and application of technologies, and to reduce the cost of producing safety-certified food. The government should also set up appropriate subsidy programs using fiscal funds to ensure the producers’ basic revenue and stimulate their enthusiasm while balancing the highest efficiency of policies and maximum social welfare by keeping the prices of safety-certified pork within a reasonable range.

## Figures and Tables

**Table 1 ijerph-15-02764-t001:** The attributes and code settings of ham butt.

Attributes	Attribute Hierarchy	Code
Certification level	(1) No certification	NOCERT
	(2) Pollution-free agricultural product certification	AGRCERT
	(3) Green food certification	GRECERT
	(4) Organic food certification	ORGCERT
Place of origin information	(1) No place of origin	NOORIGIN
	(2) With a place of origin	ORIGIN
Quality assurance mark	(1) No “no additives and veterinary drug residue label”	NOLABLE
	(2) With “no additives and veterinary drug residue label”	LABLE
Price	(1) 15 yuan/500 g	PRICE1
	(2) 25 yuan/500 g	PRICE2
	(3) 40 yuan/500 g	PRICE3

**Table 2 ijerph-15-02764-t002:** The personal characteristics statistics of the sample consumers.

Statistics Characteristics	Classification Indicator	Number of Samples (Person)	Percentage (%)
Gender	Male	371	43.96
	female	473	56.04
Age	≤30 (exclude minors)	255	30.21
	30–39	151	17.89
	40–49	223	26.42
	50–59	134	15.88
	60 or above	81	9.60
Education Level	Junior high school and below	248	29.38
	High school or secondary school	220	26.07
	College	115	13.63
	Bachelor	220	26.07
	Graduate and above	41	4.86
Annual Family Income (CNY)	50,000 yuan or less	112	13.27
	50,000–80,000 yuan	214	25.36
	80,000–100,000 yuan	256	30.33
	100,000 yuan or above	262	31.04
Is there a minor under the age of 18 at home?	Yes	413	48.93
	No	431	51.07

**Table 3 ijerph-15-02764-t003:** The consumers’ cognition and consumption of safety-certified pork.

Variable	Classification Indicator	Number	Percentage (%)
Awareness of safety-certified pork	Know very little	101	11.97
	Do not know much	376	44.55
	General	203	24.05
	Know well	149	17.65
	Very familiar with	15	1.78
Concern about the safety of pork	Never care	20	2.37
	Concern a little	106	12.56
	General	144	17.06
	Relatively pays close attention	383	45.38
	Be very concerned about	191	22.63
Trust in the quality and safety certification mark of agricultural products	Very distrustful	16	1.9
	Not trusting	170	20.1
	General	207	24.5
	More Trust	388	46.0
	Very trust	63	7.5
Is there an idea to buy safety-certified pork?	Yes	773	91.59
	No	71	8.41
Have you purchased safety-certified pork?	Yes	486	57.58
	No	358	42.42

**Table 4 ijerph-15-02764-t004:** The assigned values for the variables.

Main Effect Variable	Assigned Values for Variables	
Green food certification (GRECERT)	GRECERT = 1; AGRCERT = 0; ORGCERT = 0	
Pollution-free agricultural product certification (AGRCERT)	GRECERT = 0; AGRCERT = 1; ORGCERT = 0	
Organic food certification (ORGCERT)	GRECERT = 0; AGRCERT = 0; ORGCERT = 1	
No certification (NOCERT)	GRECERT = −1; AGRCERT = −1; ORGCERT = −1	
With “no additives and veterinary drug residue labels” (LABLE)	LABLE = 1;	
No “no additives and veterinary drug residue labels” (NOLABLE)	NOLABLE = −1	
With place of origin (ORIGIN)	ORIGIN = 1	
No place of origin (NOORIGIN)	NOORIGIN = −1	
Price	PRICE = 15; PRICE = 25; PRICE = 40	
Covariate	Assign values for variables	Mean value
Gender	Virtual variable: male = 1, female = 0	0.44
Age	Continuous variable	40.68
Education (EDU)	Continuous variable (to specific years of education)	12.81
Annual family income (INCOME)	Continuous variable (ten thousand yuan)	10.27

**Table 5 ijerph-15-02764-t005:** The regression results of the Mixed Logit model.

Variable	Estimation Coefficient	SD	95% CI	
Opinion Out (Opt Out)	−1.9257 ***	0.0482	−2.0201	−1.8313
PRICE	−0.0446 ***	0.0016	−0.0478	−0.0415
Pollution-free agricultural product certification (AGRCERT)	0.1819	0.1349	−0.0824	0.4462
Green food certification (GRECERT)	0.4111 ***	0.1394	0.1379	0.6844
Organic food certification (ORGCERT)	0.5393 ***	0.12	0.3041	0.7745
Origin information (*ORIGIN*)	0.2688 ***	0.0752	0.1215	0.4161
With “no additives and veterinary drug residue labels” (LABLE)	0.4621 ***	0.0807	0.3039	0.6203
Cross term				
Pollution-free × Gender (AGRCERT×*GENDER*)	−0.0139	0.0546	−0.1209	0.0931
Pollution-free × age (AGRCERT×*AGE*)	−0.0047	0.0252	−0.0541	0.0448
Pollution-free × Education (AGRCERT×*EDU*)	−0.0056	0.0268	−0.0581	0.0469
Pollution-free × Income (AGRCERT×*INCOME*)	0.0185	0.0271	−0.0347	0.0716
Green × Gender (GRECERT*×GENDER*)	−0.0014	0.0567	−0.1126	0.1097
Green × age (GRECERT*×AGE*)	−0.0249	0.0261	−0.076	0.0262
Green × Education (GRECERT*×EDU*)	−0.0109	0.0279	−0.0655	0.0438
Green × Income (GRECERT*×INCOME*)	0.0861 ***	0.0279	0.0313	0.1408
Organic × Gender (ORGCERT*×GENDER*)	0.0268	0.0486	−0.0685	0.1221
Organic × age (ORGCERT*×AGE*)	−0.0593 ***	0.0224	−0.1033	−0.0153
Organic × Education (ORGCERT*×EDU*)	−0.0053	0.0239	−0.0521	0.0414
Organic × Income (ORGCERT*×INCOME*)	0.0594 **	0.0241	0.0122	0.1066
Origin × Gender (*ORIGIN×GENDER*)	0.0108	0.0305	−0.0491	0.0706
Origin × age (*ORIGIN×AGE*)	−0.0245 *	0.0141	−0.0521	0.0031
Origin × Education (*ORIGIN×EDU*)	−0.013	0.015	−0.0423	0.0164
Origin × Income (*ORIGIN×INCOME*)	0.0737 ***	0.0151	0.0441	0.1033
Label × Gender (*LABEL×GENDER*)	−0.0268	0.0327	−0.0909	0.0374
Label × age (*LABEL×AGE*)	−0.0057	0.0151	−0.0353	0.0239
Label × Education (*LABEL×EDU*)	−0.0088	0.0161	−0.0403	0.0228
Label × Income (*LABEL×INCOME*)	0.0646 ***	0.0162	0.0329	0.0964
Observation sample size	844			
Log Likelihood	−8830.6405			
McFadden R^2^	0.2271			
AIC	17,715.3			

Note: ***, **, and * indicate that the estimated coefficients are significant at the statistical levels of 1%, 5%, and 10%, respectively.

**Table 6 ijerph-15-02764-t006:** The variable definition and description.

Variable Symbol	Definition
Gender	Male = 1; female = 0
Age	1–5: 1 means 30 years old and below; 2 means 30–39 years old; 3 means 40–49 years old; 4 means 50–59 years old; 5 means 60 years old or older
Edu	1–5: 1 means junior high school and junior high school; 2 means high school (including vocational high school); 3 means junior college; 4 means undergraduate; 5 means graduate student and above
Income	1–4: 1 means 50,000 yuan and below; 2 means 5–8 million yuan; 3 means 8–10 million yuan; 4 means more than 100,000 yuan
Kid	There are children under the age of 18 in the family=1; there are no children under the age of 18 in the family = 0
Understand	1–5: 1 means very unknown; 2 means less understanding; 3 means general; 4 means better understanding; 5 means very understanding
Follow	1–5: 1 means not paying attention; 2 means not paying attention; 3 means general; 4 means more attention; 5 means very concerned
Trust	1–5: 1 means very distrust; 2 means less trust; 3 means general; 4 means compare trust; 5 means very trust
Y1	Willingness to buy safely certified pork: willing = 1; unwilling = 2
Y2	Purchase safety-certified pork: purchase = 1; no purchase = 0

**Table 7 ijerph-15-02764-t007:** The Binary Logit regression model estimation results for consumers purchasing safety-certified pork.

Variables	B	Wals	Exp (B)	EXP (B) 95% CI	
				Lower Limit	Upper Limit
Gender	−0.536 ***	11.165	0.585	0.427	0.801
Age	0.162 **	4.318	1.176	1.009	1.370
Education	0.041	0.265	1.042	0.891	1.218
Family annual income	0.138 *	2.947	1.148	0.981	1.344
Is there a child under the age of 18 at home?	−0.065	0.163	0.937	0.683	1.285
Trust in the quality and safety certification mark of agricultural products	0.227 ***	6.743	1.255	1.057	1.489
Awareness of safety-certified pork	0.614 ***	43.692	1.848	1.541	2.218
Concern about the quality and safety of pork	0.138 *	2.917	1.148	0.980	1.345

Note: ***, **, and * indicate that the correlation is significant at the statistical levels of 1%, 5%, and 10%, respectively.

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
