# Peer review of "Research on the Irrational Behavior of Consumers’ Safe Consumption and Its Influencing Factors"

_ijerph, 2018, doi:10.3390/ijerph15122764_

Round 1
Reviewer 1 Report
The manuscript entitled “Research on Irrational Equilibrium of Consumers’ Safe Consumption and Its Influencing Factors” presents issues associated with food safety.
General comments:
-Authors should carefully check and correct the typos (e.g. line 15, 17, 38, 249, 253, etc. )
-The term “irrational equilibrium“ is not well support by the authors. It will be better if authors replace it by e.g. simple “irrational behavior” (as it is referred in the literature background)
-The title does not reflect the findings
-The language used by authors was improved but in many paragraphs it is still unprofessional or unscientific (e.g. “hotspot issue”; “we chose pork”, “Therefore, the irrational behavior of consumer safe consumption will have a certain impact on the safe food market”, “This paper selects pork as a typical safety…”). The manuscript should be corrected by native English speaker (familiar with scientific wording) or maybe by the professional agency.
Introduction:
-Lines 43-54 – references are required.
-The proper definition of safe and unsafe food must be presented. What did authors mean by “consumers […] may not actually buy safe food”. On the regular (legal) market in developed countries all food products must be sold as a safe ones (this is regulated by the law). Therefore it is difficult to understand the concept of this paragraphs. It is well known that some consumers are afraid of meat products due to some scandals (e.g. BSM, ASF, etc.) but the meat products are generally safe and nobody sells unsafe food products. This concept must be presented by authors in more reasonable way. The scale of fraud on the food market in China must be presented in numbers (if it is a big problem). Maybe authors have in mind the informal market (if so it must be presented properly).
-Lines 69-71 – It is not true. Irrational consumption means “not based on logical reasons or clear thinking” but not “consumers consume without even thinking or with few knowledge and understanding of the product.”
-Lines 77-78 – references are required.
-Line 79 – references are required. Please present the scale of this phenomena
-Authors should rewrite this section because it is difficult to follow what do authors mean. In line 81, the term “green food” is presented, but it is difficult to judge why it is important for the subject (in this paragraph).
-Lines 100-103 – the example (organic beef) is opposite to the “organic tea” case (lines 85-86) – the difference must be explained.
-Line 113 – It should be “researches” instead of “Domestic and foreign researches”
-Lines 114-115 – “…. affecting consumer safe consumption” – this sentence is awkward
Experimental Design and Sample Analysis:
-Authors presented the information that pre-survey was conducted, but the information about accuracy, reliability consistency of this questionnaire (validation) are still missing.
-Table 2 – the income could be presented in international currency (additionally).
-Tables 3 – use “number” instead of “samples size (person)”
-Tables 3 – question “Awareness of safety certified pork“ – the translation of the scale seems to be inadequate (the scale is not continuous in the term – term “general” does not fit). Similar situation is observed in the case of other questions.
-For question “Is there an idea to buy safe certified pork?” is it for the general idea or is it dedicated to the specific consumer/respondent. I think that there was a problem with the translation of meaning. Please check it.
-Statistical analysis section must be presented – what statistical tests were used for analysis
-For the research that involves human subjects the rules of the Declaration of Helsinki of 1975 must be applied (even for on-line survey), including ethics commission approval and especially informed consent. Please add the information about number of ethics commission approval.
Conclusions and Discussion:
-The title of this section should be “Discussion and Conclusions”
-There is no discussion section. Authors could move some paragraphs from introduction (lines 111-133). This section must be totally rewritten. Authors should relate the findings to those of similar studies and point the differences and similarities between the studies. Authors should add the appropriate references in this section.
-Conclusion must emphasize the findings and the novelty! In the present form there is an insufficient novelty.
Author Response
Thank you for reading and commenting on the paper. We have re-adjusted the paper again. The specific modification are as follows.
Comments and Suggestions for Authors
The manuscript entitled “Research on Irrational Equilibrium of Consumers’ Safe Consumption and Its Influencing Factors” presents issues associated with food safety.
General comments:
-Authors should carefully check and correct the typos (e.g. line 15, 17, 38, 249, 253, etc. )
Response:
In line15, we added a space after the word “pork”.
In line 17, we added a space after the word “RPL”.
In line 38, we added a space after the word “water”.
In line 244, we replaced “pork buying experience” with “pork purchase experience”.
In line 248,we didn’t find the typo, so we keep the original sentence.
-The term “irrational equilibrium“ is not well support by the authors. It will be better if authors replace it by e.g. simple “irrational behavior” (as it is referred in the literature background)
-The title does not reflect the findings
Response:
We adjusted our paper title “Research on Irrational Equilibrium of Consumers’ Safe Consumption and Its Influencing Factors” to “Research on Irrational Behavior of Consumers’ Safe Consumption and Its Influencing Factors” in line 2-3.The word “irrational equilibrium“ was adjusted in the corresponding section of the article.
-The language used by authors was improved but in many paragraphs it is still unprofessional or unscientific (e.g. “hotspot issue”; “we chose pork”, “Therefore, the irrational behavior of consumer safe consumption will have a certain impact on the safe food market”, “This paper selects pork as a typical safety…”). The manuscript should be corrected by native English speaker (familiar with scientific wording) or maybe by the professional agency.
Response:
With the help of a native English speaker, we reorganize sentences as follows:
In line 15, we replaced “we chose pork” with “we chose to use pork”.
In line 38, we replaced “hotspot issue” with “hot topic”
In line 76, we reorganized the sentence as follows “Therefore, the irrational behavior of consumer’s safe consumption will have a certain impact on the safe food market.”
In line 385, we reorganized the sentence as follows “This paper studies consumers’ willingness to consume and purchasing behavior of certified pork which is a typical safety-certified agricultural product”
Introduction:
-Lines 43-54 – references are required.
Response:
The sentence in line 43 and in line 45, we added references separately.
-The proper definition of safe and unsafe food must be presented. What did authors mean by “consumers […] may not actually buy safe food”. On the regular (legal) market in (this is regulated by the law). Therefore it is difficult to understand the concept of this paragraphs. It is well known that some consumers are afraid of meat products due to some scandals (e.g. BSM, ASF, etc.) but the meat products are generally safe and nobody sells unsafe food products. This concept must be presented by authors in more reasonable way. The scale of fraud on the food market in China must be presented in numbers (if it is a big problem). Maybe authors have in mind the informal market (if so it must be presented properly).
Response:
China is still a developing country, and the market mechanism is still not perfect (the government’s supervision and punishment system is yet to be perfect). Therefore, producers will have some speculative behaviors by seizing market loopholes and circulating unsafe pork to the market. Therefore, there are still a certain number of unqualified pork in the Chinese market, such as sick pig pork. And we give an example in line 47 to explain this phenomenon.
And the definition of safe food, we defined in line 42 that safe food should be a product that does not have the potential to harm or threaten human health, including the safety of production and operations, the resulting and process, as well as the reality and future.
-Lines 69-71 – It is not true. Irrational consumption means “not based on logical reasons or clear thinking” but not “consumers consume without even thinking or with few knowledge and understanding of the product.”
Response:
This error was corrected in line 72 as follow “Irrational consumption generally occurs in the situation when consumers consume without logical reasons or clear thinking due to few knowledge and understanding of the product.”
-Lines 77-78 – references are required.
-Line 79 – references are required. Please present the scale of this phenomena
-Authors should rewrite this section because it is difficult to follow what do authors mean. In line 81, the term “green food” is presented, but it is difficult to judge why it is important for the subject (in this paragraph).
Response:
The reference for line 79-80 have been added. And the sentence in line 82 is associated with the next sentence. In China, “green food” is one of food quality and the safety standards, so we chose this example to proof the view “there is still lack of trust of Chinese consumers on food quality and the safety standards”.
-Lines 100-103 – the example (organic beef) is opposite to the “organic tea” case (lines 85-86) – the difference must be explained.
Response:
In the case of organic tea, consumers are willing to buy products with organic signs, but consumers are only willing to pay a premium of 39.96% for organic tea ,while the market price of organic tea generally has a premium of more than 50% when compared with ordinary tea. This is not inconsistent with the case of organic beef. Chinese consumers had a significant preference for both organic certified tea and beef.
-Line 113 – It should be “researches” instead of “Domestic and foreign researches”
-Lines 114-115 – “…. affecting consumer safe consumption” – this sentence is awkward
Response:
The “Domestic and foreign researches” have been replaced with “researches” in line 115, and the sentence in line 115-116 have been reorganized.
Experimental Design and Sample Analysis:
-Authors presented the information that pre-survey was conducted, but the information about accuracy, reliability consistency of this questionnaire (validation) are still missing.
Response:
In response to this question, we added some details in the line 235. In order to ensure accuracy, reliability consistency of this questionnaire, we designed a pre-survey and summarize and improve the quality of the questionnaire through expert evaluation and internal test, and all of our enumerators had been trained to familiarize themselves with the questionnaire. And in order to ensure the reasonable distribution of the samples and the authenticity and effectiveness of the data, this survey is conducted following the principle of layered design.
-Table 2 – the income could be presented in international currency (additionally).
-Tables 3 – use “number” instead of “samples size (person)”
-Tables 3 – question “Awareness of safety certified pork“ – the translation of the scale seems to be inadequate (the scale is not continuous in the term – term “general” does not fit). Similar situation is observed in the case of other questions.
-For question “Is there an idea to buy safe certified pork?” is it for the general idea or is it dedicated to the specific consumer/respondent. I think that there was a problem with the translation of meaning. Please check it.
-Statistical analysis section must be presented – what statistical tests were used for analysis
Response:
Some details have been adjusted in the table 2 and table 3.
For the description of the question, we have referenced some of the literature, and many literature use this form of expression, so we retain the original sentence.
For question “Is there an idea to buy safe certified pork?”, our investigate selected consumers with pork purchase experience as respondents (in line 244). So it is dedicated to the specific consumer.
Statistical analysis section have been added in line 254 and line 277.
-For the research that involves human subjects the rules of the Declaration of Helsinki of 1975 must be applied (even for on-line survey), including ethics commission approval and especially informed consent. Please add the information about number of ethics commission approval.
Response:
We are very grateful to the review for raising questions about the ethics commission, and we had replied to this question before. We consulted the relevant institutions, they judged that our methods of social investigation did not touch on the human moral level, so there is no ethics commission approval.
Conclusions and Discussion:
-The title of this section should be “Discussion and Conclusions”
-There is no discussion section. Authors could move some paragraphs from introduction (lines 111-133). This section must be totally rewritten. Authors should relate the findings to those of similar studies and point the differences and similarities between the studies. Authors should add the appropriate references in this section.
-Conclusion must emphasize the findings and the novelty! In the present form there is an insufficient novelty.
Response:
The title of this section have been adjusted to “Discussion and Conclusions”. And we reorganized this section.
Reviewer 2 Report
Generally, the writing is quite clear. I would recommend "accept" after minor revision.
(1) The introduction section is quite long. I would suggest that the authors may use two or three paragraphs;
(2) The authors should clearly point out the research objective and research contributions in the introduction section;
(3) The authors may discuss the future research directions in the last part of Section 6.
Author Response
Thank you for reading and commenting on the paper. We have re-adjusted the paper again. The specific modification are as follows.
Comments and Suggestions for Authors
Generally, the writing is quite clear. I would recommend "accept" after minor revision.
(1) The introduction section is quite long. I would suggest that the authors may use two or three paragraphs;
Response:
We divided the introduction into 2 paragraphs for description.
(2) The authors should clearly point out the research objective and research contributions in the introduction section;
Response:
The research objective and research contributions is placed at the end of the introduction from line 61-66.
(3) The authors may discuss the future research directions in the last part of Section 6.
Response:
As for the future research directions, we think foods of other categories and data from other provinces in China, especially in the western region, can be tapped to further complement and improve existing research results, we placed this content in line 400-407.
Reviewer 3 Report
In the assessment of the paper submitted for the review, I specifically focused on the discussed issues, applied research procedure, substantive content of the paper and its structure.
The considerations conducted in the paper are focused on such categories as: safe consumption, irrational equilibrium, safety-certified pork, consumer preference, purchase intention. The subject area discussed in the paper should be considered important and topical.
The structure of the paper is clear. The value of the paper results from appropriate combination of literature studies with the results of an empirical research, which was conducted on a sample of 844 respondents (consumers in Jiangsu Province and Anhui Province). Presentation of the results of own empirical research is the value of the paper. However, deliberations conducted in the paper need to be expanded. Therefore, it is specifically recommended to:
- describe the limitations of conducted research and to indicate the trends for further empirical research (as a separate part of article),
- specify the managerial implications (as a separate part of article),
-indicate the time the empirical research was conducted (in abstract).
Author Response
Thank you for reading and commenting on the paper. We have re-adjusted the paper again. The specific modification are as follows.
Comments and Suggestions for Authors
In the assessment of the paper submitted for the review, I specifically focused on the discussed issues, applied research procedure, substantive content of the paper and its structure.
The considerations conducted in the paper are focused on such categories as: safe consumption, irrational equilibrium, safety-certified pork, consumer preference, purchase intention. The subject area discussed in the paper should be considered important and topical.
The structure of the paper is clear. The value of the paper results from appropriate combination of literature studies with the results of an empirical research, which was conducted on a sample of 844 respondents (consumers in Jiangsu Province and Anhui Province). Presentation of the results of own empirical research is the value of the paper. However, deliberations conducted in the paper need to be expanded. Therefore, it is specifically recommended to:
- describe the limitations of conducted research and to indicate the trends for further empirical research (as a separate part of article),
Response:
Accounting for the largest proportion of meat consumption in China and frequently pork safety issues, this study adopts safety certified pork as the experimental subject to explore the influencing factors of consumers’ safe consumption and preferences over product safety attribute hierarchy. In view of the certain differences between different safety certification product categories, and each category has its own particularity, in the future research, foods of other categories can be tapped to further complement and improve existing research results.
Due to the convenience and geographical advantages, the survey locations were set up in Jiangsu Province and Anhui Province. Therefore, in future research, data from other provinces in China can be tapped, especially in the western region.
We put this above in line 400-407 as a separate part of article in the section of Discussion and Conclusions.
- specify the managerial implications (as a separate part of article),
Response:
For the managerial implications,the paper gives recommendations on enhancing consumer trust in food safety and improving producers' enthusiasm for safe food production, which can be found in line 408-422 as a separate part.
-indicate the time the empirical research was conducted (in abstract).
Response:
We added the time the empirical research in abstract (in line 17).
Round 2
Reviewer 2 Report
Overall, I am happy with the revision works that the authors have conducted. Thus, I recommend that the paper should be accepted.
Reviewer 3 Report
Thank You for introducing my suggestions in article.